# Differential inhibition onto developing and mature granule cells generates high-frequency filters with variable gain

María Belén Pardi[1,2], Mora Belén Ogando[1], Alejandro F Schinder[2]*, Antonia Marin-Burgin[1]*

[1]Instituto de Investigación en Biomedicina de Buenos Aires—CONICET—Partner Institute of the Max Planck Society, Buenos Aires, Argentina; [2]Laboratory of Neuronal Plasticity, Leloir Institute—CONICET, Buenos Aires, Argentina

**Abstract** Adult hippocampal neurogenesis provides the dentate gyrus with heterogeneous populations of granule cells (GC) originated at different times. The contribution of these cells to information encoding is under current investigation. Here, we show that incoming spike trains activate different populations of GC determined by the stimulation frequency and GC age. Immature GC respond to a wider range of stimulus frequencies, whereas mature GC are less responsive at high frequencies. This difference is dictated by feedforward inhibition, which restricts mature GC activation. Yet, the stronger inhibition of mature GC results in a higher temporal fidelity compared to that of immature GC. Thus, hippocampal inputs activate two populations of neurons with variable frequency filters: immature cells, with wide-range responses, that are reliable transmitters of the incoming frequency, and mature neurons, with narrow frequency response, that are precise at informing the beginning of the stimulus, but with a sparse activity.

*For correspondence:
aschinder@leloir.org.ar (AFS);
aburgin@ibioba-mpsp-conicet.
gov.ar (AM-B)

Competing interests: The authors declare that no competing interests exist.

## Introduction

The dentate gyrus (DG) is the main entrance of information to the hippocampus (*Andersen et al., 1971*). Activity arriving to the hippocampus from the entorhinal cortex (EC) should be processed by heterogeneous populations of granule cells (GC) of different stages of maturation (*Zhao et al., 2008*; *Lepousez et al., 2015*). Neural progenitor cells of the adult subgranular zone give rise to dentate GC that develop and mature over several weeks. Developing GC follow a precise sequence to establish their afferent connectivity and functional maturation. New born neurons are initially contacted by dendritic GABAergic terminals, followed by glutamatergic axons and last by GABAergic perisomatic contacts. In parallel, their membrane resistance decreases and excitability becomes mature (*Espósito et al., 2005*; *Ge et al., 2006*). When fully developed, adult-born neurons achieve a functional profile that is indistinguishable from that of all other GC as reflected by their inputs, intrinsic membrane properties, and firing behavior (*Laplagne et al., 2006*, *2007*). In addition, adult-born GC form functional glutamatergic synapses onto DG interneurons and CA3 pyramidal cells (*Toni et al., 2008*; *Temprana et al., 2015*), indicating that new neurons receive, process, and convey information onto target neurons and participate in hippocampal function.

As activity arrives to the DG, it does not only activate GC but also inhibitory circuits (*Buzsaki, 1984*). The interaction of the excitatory activity arriving from EC with the recruited inhibition will ultimately determine GC's activity. Inhibitory circuits can have a profound impact in the processing of afferent activity (*Markwardt et al., 2009*; *Isaacson and Scanziani, 2011*; *Ikrar et al., 2013*); for example, they can ensure a temporal fidelity of responses (*Pouille and Scanziani, 2001*) and also modulate neuronal firing in response to afferent inputs (*Pouille et al., 2009*; *Dieni et al., 2013*).

**eLife digest** A number of cell types in the body are capable of dividing to produce two new cells. These are used either to replace damaged or worn out tissue, or to satisfy a need for additional cells. By contrast, the vast majority of the billions of neurons in the brain are produced before birth, and only a handful of brain regions retain the ability to generate new neurons throughout life.

One of these brain regions is the hippocampus, which has roles in memory and navigation. The zone of the hippocampus that receives signals from other parts of the brain—known as the dentate gyrus—contains cells called granule cells that are still able to divide in adult animals. Newly formed cells mature over the course of a few weeks, but whether immature and mature cells make different contributions to processing the information that enters the hippocampus is unclear.

By stimulating slices of mouse hippocampus using sequences of electrical pulses similar to those that occur naturally in the brain, Pardi et al. have shown that the granule cells' responses vary with age. Both mature and 4-week-old granule cells responded more strongly when the electrical pulses were applied at a slower rate. However, the immature cells could also respond to pulses applied at a faster rate more reliably than their mature counterparts.

In contrast, mature cells signaled the arrival of the first pulse in a sequence more precisely than immature cells, with the majority of mature granule cells firing within the first three milliseconds of receiving a pulse. Additional experiments revealed that these differences arise because mature cells are more easily prevented from firing by inhibitory neurons that contact them, particularly in the presence of rapid sequences of pulses.

By using immature and mature granule cells to encode different aspects of an incoming stimulus, the hippocampus may thus be able to maximize its ability to process information. These results raise questions regarding how parallel processing by immature and mature granule cells could affect the processing in CA3, the area that receives the information sent by these cells.

We have previously demonstrated that due to a smaller and slower inhibition, immature GC are preferentially recruited in response to a single stimulation of its afferent entorhinal pathway (*Marín-Burgin et al., 2012*). However, activity arriving to the hippocampus does not only come as a single stimulus but varies during different brain states in frequencies ranging from theta to gamma (*Buzsáki and Draguhn, 2004*; *Buzsáki and Moser, 2013*). How is this information transformed into activity of the granule cells? Are immature and mature populations of neurons differentially recruited at different frequencies? If that was the case, information arriving to DG could be differentially channeled into activation of different populations of granule cells depending on the frequency.

In this work, we have addressed these questions focusing on the understanding of how excitation and inhibition interact to determine the response of GC. We found that due to a weaker influence of feedforward inhibitory circuits, immature neurons can respond to a wider range of frequencies than mature neurons, however, they are less time locked to the incoming stimuli. Mature neurons on the contrary show a higher temporal fidelity but are less effective in their responses to frequency. We suggest that immature and mature GC reflect complementary aspects of incoming activity.

## Results

### Activation of mature and immature granule cells at different frequencies

We investigated how immature GC process afferent activity from entorhinal inputs arriving at different frequencies and how they compare to mature GC in the adult mice hippocampus. We selected 4-week-old neurons as immature GC because this is the earliest stage at which adult-born GC can be reliably activated by an excitatory drive (*Mongiat et al., 2009*), exhibit funtional properties that distinguish them from mature neurons (*Wang et al., 2000*; *Snyder et al., 2001*; *Schmidt-Hieber et al., 2004*; *Espósito et al., 2005*; *Ge et al., 2007*; *Marín-Burgin et al., 2012*), and are already connected with postsynaptic targets (*Gu et al., 2012*; *Marín-Burgin et al., 2012*; *Temprana et al., 2015*). Adult-born GC were retrovirally labeled to express RFP or GFP and acute hippocampal slices were prepared 4 weeks post retroviral injection (4 wpi) (*Figure 1A*).

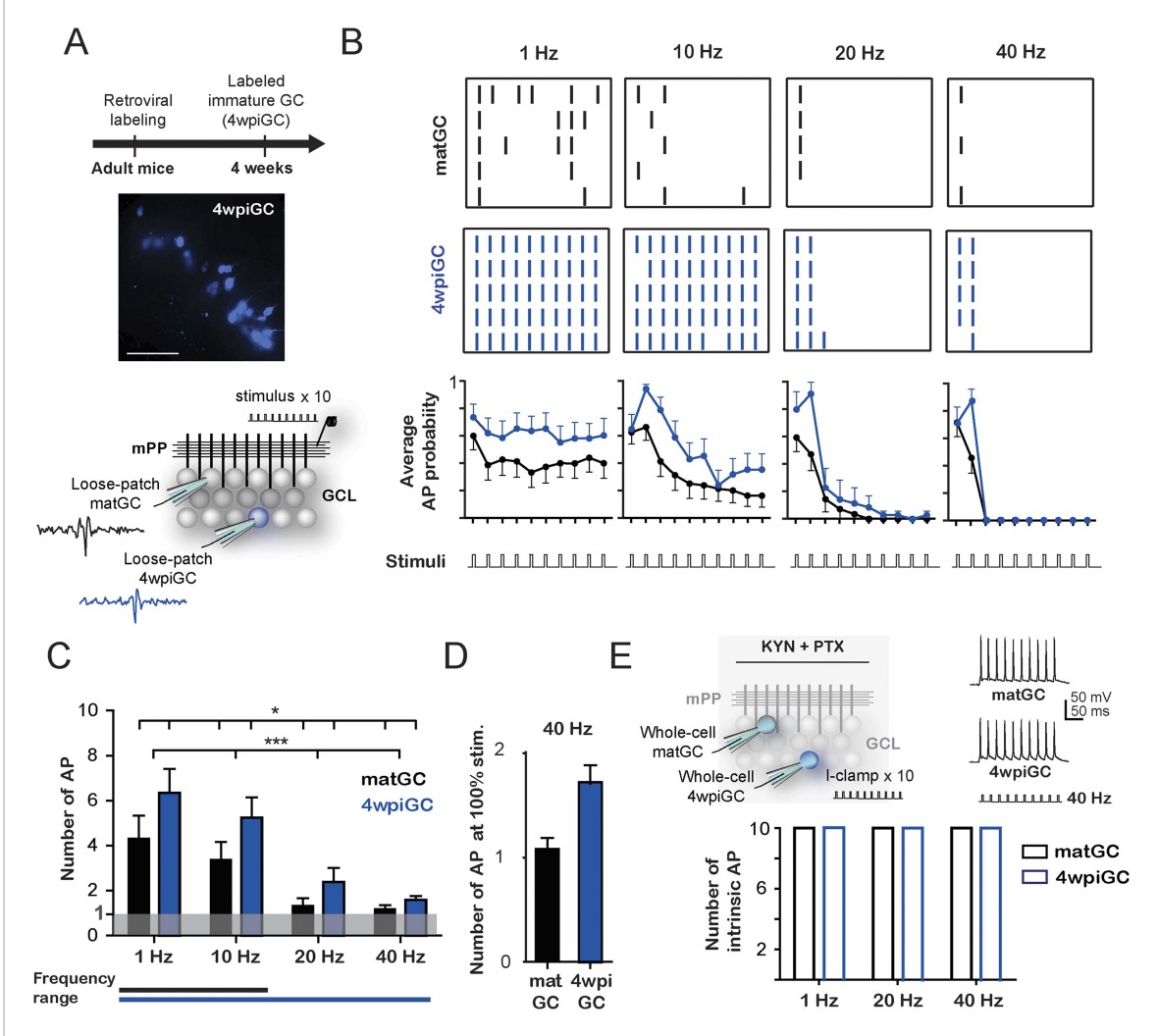

Figure 1. Frequency-dependent activation of mature and immature granule cells. (A) The image shows a hippocampus slice with 4-weeks-old GC (4wpiGC) expressing RFP (pseudo-colored in blue). Scale bar: 50 µm. The upper timeline indicates the time of retroviral injection. The lower scheme shows the recording configuration: a stimulating electrode was placed in the medial perforant path (mPP) to deliver 10 stimuli at different frequencies; the stimulation intensity was kept at 50% fEPSP. Loose patch recordings were obtained from mature GC (matGC) and 4wpiGC to detect spikes. (B) Raster plots from one matGC (black) and one 4wpiGC (blue) at 1 Hz, 10 Hz, 20 Hz, and 40 Hz. Each color dash denotes a spike. Columns: stimulation pulses at the four frequencies. Rows: stimulation trains. Lower panels are the average action potential probability at each pulse of the train for all the data from matGC (black) and 4wpiGC (blue). (C) Average of the sum of action potentials evoked by stimulation trains of 1 Hz, 10 Hz, 20 Hz, and 40 Hz, in matGC (black bars) and 4wpiGC (blue bars). Activation decreases with frequency in both cells and is higher in 4wpiGC than in matGC at all stimulation frequencies (two-way ANOVA, variation between GC: *p < 0.05; variation in frequency: ***p < 0.001; interaction: ns, p > 0.05). N (4wpiGC) = 12, 13, 7, and 11 cells and N (matGC) = 15, 16, 11, 14 cells for 1 Hz, 10 Hz, 20 Hz, and 40 Hz, respectively. The frequency range lines at the bottom shows the range of frequencies that GC responded with a number of action potentials significantly different from 1 (Wilcoxon signed-rank test, at 20 and 40 Hz, p < 0.05 for 4wpiGC, p > 0.05 for matGC). (D) Average number of action potentials when stimulus was at 100% spiking of each cell after the first stimulation pulse (above threshold) in matGC (black) and 4wpiGC (blue) at 40 Hz stimulation. (E) Upper scheme shows the recording configuration. Whole cell current clamp (I-Clamp) recordings were obtained from matGC and 4wpiGC. Synaptic activity was blocked by kynurenic acid (KYN) and picrotoxin (PTX). Intrinsic spiking activity was evaluated by injecting 10 depolarizing brief square current pulses at an intensity that evoked action potentials in the 10 pulses at 1 Hz. Cells were tested at frequencies every 20 Hz. The lower graph shows the mean number of action potentials evoked at 1 Hz, 20 Hz, and 40 Hz. All recorded cells in whole-cell fired 10 action potentials when 10 current pulses were delivered at these frequencies (N = 4 cells for both GC). Error bars indicate SEM.

The following figure supplements are available for figure 1:

Figure supplement 1. Input normalization.

Figure supplement 2. Immature GC are more effective at responding to frequency than matGC.

To investigate how 4-week-old (4wpiGC) and mature (matGC) GC respond to conditions of activity that resemble those occurring during hippocampus-dependent behaviors (*Leutgeb et al., 2007*), we measured spiking in response to trains of different frequencies delivered to the medial perforant pathway (mPP). We used electrophysiological loose patch recordings to characterize the activation profile of GC at the single cell level (*Figure 1A*). The intensity of the stimulus delivered to mPP was normalized using field recordings to compare physiologically equivalent inputs across slices and a fixed stimulation intensity (50% fEPSP$_{slope}$) was chosen for all experiments (*Figure 1—figure supplement 1*). Five trains of 10 pulses were delivered at 1, 10, 20, and 40 Hz for each cell (*Figure 1B*).

Immature GC activate with a higher probability than mature GC at all frequencies, yet both GC populations showed decreased responsiveness for trains delivered at increasing frequency (*Figure 1B*). Activation of mature and immature GC decreased within each train, and this reduction was more pronounced for the higher frequencies (*Figure 1B*). To quantitatively compare activation between mature and immature GC across frequencies, we calculated the average spike number for all 10 pulses for each frequency. Results indicate that the activation of GC decreases drastically with frequency. However, immature GC spiked more reliably than mature GC at all frequencies (*Figure 1C*). Stimulation at 1 Hz represents a situation similar to applying a single pulse; accordingly, spiking probability in matGC was around 50% (5 spikes average), consistent with the responses evoked using stimulation intensities of about 50% of the fEPSP and with previous results (*Marín-Burgin et al., 2012*). While 4wpiGC significantly responded with more than 1 spike to all frequencies, matGC only responded with more than 1 spike to 1 and 10 Hz, indicating that matGC could not be synaptically activated to fire at 20 and 40 Hz (*Figure 1C*). Furthermore, matGC could not spike two consecutive action potentials even if the stimulation intensity was chosen to record neurons above threshold (*Figure 1D*). The higher activation levels of 4wpiGC at all frequencies indicate that they could reproduce a wider range of frequencies than matGC. These data indicate that dentate GC have frequency filters with variable gain depending on their age.

The ability of GC to spike at a given frequency reflects its ability to re-transmit time information to their target neurons. To analyze this aspect, we calculated the efficacy to reproduce each frequency. This was computed as the average of the probability of occurrence of action potentials in the same frequency range as that given by the stimulus. The results show that 4wpiGC have higher efficacy than matGC at all frequencies (*Figure 1—figure supplement 2*). This observation indicates that the population of immature GC has a higher capability to retransmit temporal information to their postsynaptic targets than the population of mature neurons. However, 4wpiGC and matGC display lower spiking efficacy at higher frequencies, indicating that both neuronal populations act as low pass filters.

The decreased activation with frequency could simply be due to intrinsic physiological restrictions of GC or instead to characteristics of the activated circuit. Therefore, we studied the intrinsic capacity of 4wpiGC and matGC to fire action potentials at 20 and 40 Hz in response to injection of current pulses. Whole-cell current-clamp recordings were obtained from 4wpiGC and matGC, and trains of square current pulses were injected at increasing frequencies. All recorded GC started showing failures at frequencies near 100 Hz. Importantly, both 4wpiGC and matGC consistently followed 20 and 40 Hz (*Figure 1E*). These results indicate that the decrease in GC' activation observed when the mPP was stimulated with increasing frequencies was not due to an intrinsic incapacity of GC to spike but, rather, to properties of the activated circuit.

## Inhibitory control of activity in mature and immature GC

To understand the mechanisms underlying the differential filtering found in matGC vs 4wpiGC, we studied the individual circuit components. Activation of mPP axons not only produces monosynaptic glutamatergic excitation of GC but also recruits feedforward GABAergic inhibitory circuits, which can modulate neuronal firing in response to afferent inputs (*Buzsaki and Eidelberg, 1981*; *Pouille et al., 2009*; *Ewell and Jones, 2010*; *Marín-Burgin et al., 2012*). Thus, we evaluated the effect of blocking GABAergic inhibition with picrotoxin (PTX) in the response of 4wpiGC and matGC to stimulation of the mPP with 10 pulses at 1, 10, 20, and 40 Hz (*Figure 2A*). Notably, in the absence of inhibition activation of 4wpiGC vs matGC was similar at all frequencies (*Figure 2B*). This observation indicates that the inhibitory circuit is responsible for generating activation differences among 4wpiGC and matGC after train stimulation of the mPP.

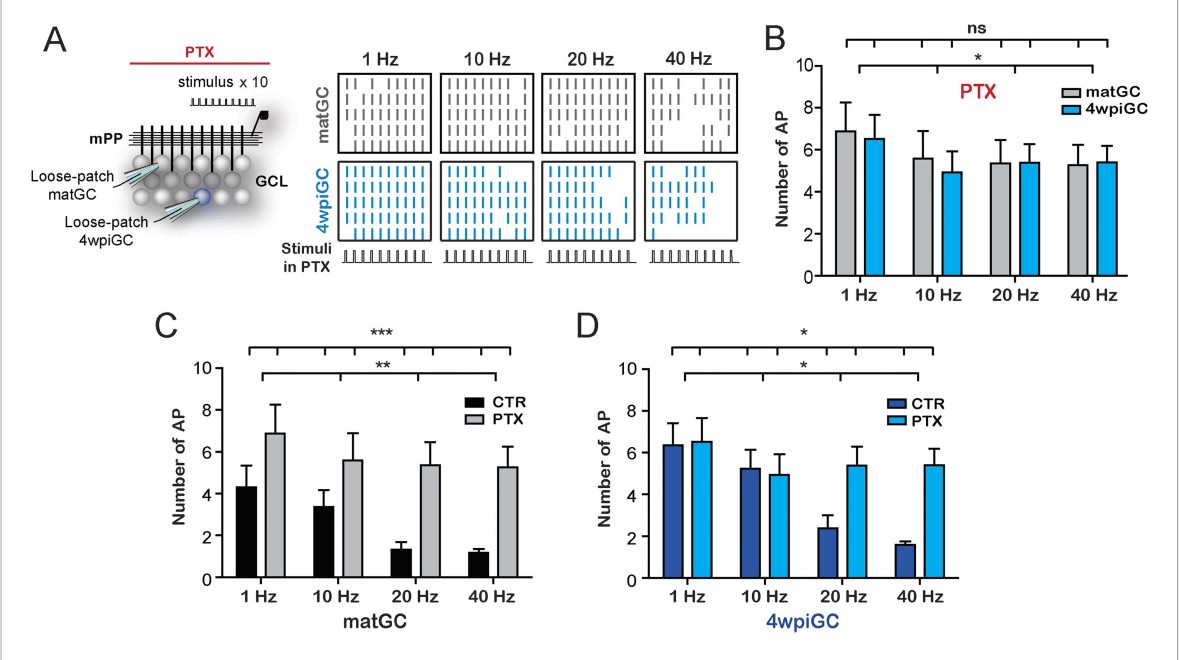

**Figure 2**. GABAergic inhibition generates the difference between mature and immature GC. (**A**) Left, the scheme shows the recording configuration, a stimulating electrode was placed in the medial perforant path (mPP) to deliver 10 stimuli at different frequencies; the stimulation intensity was kept at 50% fEPSP. Loose patch recordings were obtained from mature GC (matGC) and 4wpiGC to detect spikes in the presence of picrotoxin (PTX). Right, Raster plots from one matGC (in black) and one 4wpiGC (in blue) at 1 Hz, 10 Hz, 20 Hz, and 40 Hz. Each color line denotes a spike. Responses of neurons were recorded in PTX. (**B**) Average of the sum of action potentials evoked by stimulation trains of 1 Hz, 10 Hz, 20 Hz, and 40 Hz, in matGC (gray columns) and 4wpiGC (light blue columns) in the presence of PTX. Activation slightly decreases with frequency in both cells, but there were not significant differences among matGC and 4wpiGC (two-way ANOVA, variation between GC: ns; variation in frequency: *p < 0.05; interaction: ns, p > 0.05 N = 23 cells for both GC at the four frequencies). (**C**) Comparison of matGC activation to stimulation with trains in control conditions and when inhibition was blocked with PTX. Activation of matGC was higher in the presence of PTX at all frequencies (two-way ANOVA, variation between treatments: ***p < 0.001; frequency variation. **p < 0.01; interaction: ns, p > 0.05). (**D**) Comparison of 4wpiGC activation to stimulation with trains in control conditions and when inhibition was blocked with PTX. The effect of blocking inhibition in 4wpiGC varies with frequency. Activation increases only at high stimulation frequencies (two-way ANOVA variation between treatments: *p < 0.05; variation between frequencies: *p < 0.05; positive interaction p < 0.01; at 40 Hz, p < 0.01, Bonferroni post-test). Error bars indicate SEM.

Activation of matGC is strongly controlled by inhibition, since their activity significantly increases with PTX at every frequency (*Figure 2C*). Interestingly, activation levels in 4wpiGC did not change with the blockage of inhibition at 1 and 10 Hz, indicating that inhibitory circuits do not control activity in immature neurons when stimuli arrive every 100 ms or more. At 20 and 40 Hz however, when the separation of stimuli is 50 ms or less, the recruitment of inhibitory circuits does restrict the response of immature neurons (*Figure 2D*). These results are extremely interesting because they indicate that there is a different temporal window in which inhibition restricts mature and immature GC activation. Immature neurons have a wider time window during which excitation can elicit spikes without being affected by inhibition. The differential effect of inhibition on immature and mature GC explains the wider range of responses of immature GC to stimuli arriving at frequencies ranging from 1 to 40 Hz, as compared to mature neurons which rarely reproduce high frequencies (*Figure 1C*), demonstrating that each population have different gains to filter frequency. In addition, the results reveal that inhibitory circuits generate a wide variability of responses in the population of GC, since not only are they responsible for the differences in activation between 4wpiGC and matGC, but also for the frequency dependence of the activation.

## Synaptic mechanisms underlying frequency dependence and responsiveness of immature and mature GC

To study how the recruited inhibition affects the activation profiles of 4wpiGC and matGC, we investigated the precise contribution of excitatory and inhibitory components that control the

activation of GC at different frequencies. We performed whole-cell voltage-clamp recordings to measure excitatory and inhibitory responses elicited at each stimulation frequency at a fix intensity as wherein before. Activation of mPP produced excitatory postsynaptic currents (EPSC) and inhibitory postsynaptic currents (IPSC), indicating that glutamatergic entorhinal axons directly activate immature and mature GC and also recruit inhibitory interneurons that synapse onto both populations (*Figure 3A*). In order to assess the contribution of the IPSC and EPSC to the activation of GC, we analyzed the excitatory postsynaptic conductance (EPSG) and inhibitory postsynaptic conductance (IPSG) at a time close to the generation time of an action potential (*Figure 3A*). Comparison of latencies of evoked action potentials and EPSC peaks show that these events occur close in time (*Figure 3B*). Therefore, we quantified the peak EPSG and IPSG value at the peak of the EPSC at all frequencies (*Figure 3A–E*).

Stimulation of the mPP evokes slower IPSC in 4wpiGC than in matGC (*Figure 3C*), and as a result inhibition around action potential time is weak in 4wpiGC at frequencies were stimuli are at least 100 ms apart (at 1 Hz and 10 Hz) (*Figure 3D*). The slower IPSC rising kinetics results in a delayed inhibition that poorly influences spiking of 4wpiGC at low frequencies. IPSG evoked at 1 to 10 Hz is maintained under EPSG values in both matGC and 4wpiGC. However, IPSG values in immature neurons are very small, resulting in a higher excitation to inhibition balance than that of mature neurons, which explains their higher spiking probability (*Figure 3—figure supplement 1*). At 20 and 40 Hz, IPSG values increase over EPSG along the stimulation train, restricting the ability of the neurons to spike (*Figure 3D* and *Figure 3—figure supplement 1*). Interestingly, in 4wpiGC the IPSG exceeds the EPSG later in the train compared to matGC, which might explain the higher efficacy of 4wpiGC to transmit spike trains within the first pulses of the train.

To make quantitative comparisons between the currents evoked in 4wpiGC and matGC at different frequencies, we calculated the mean peak EPSG, and the mean IPSG value at the peak EPSG, for the 10 pulses at each frequency. Thus, we obtained a single value of the EPSG and a single value of the IPSG, for 4wpiGC and matGC at each frequency. These data show that the evoked inhibition in both populations of GC increases with frequency. Excitation, on the contrary, slightly decreases (*Figure 3E*). This slight decrease in the total excitation with the frequency explains the slight decrease in the activation of GC when inhibition was blocked (*Figure 2B*). On the other hand, the large increase in inhibition with frequency, explains the large differences found in the activation of GC among frequencies in the conditions in which the circuit was intact (*Figure 1C*). Notably, inhibition particularly increases at 20 Hz and 40 Hz, and this coincides with the greatest decrease in the activation of GC. Both excitatory and inhibitory conductances are smaller in 4wpiGC than in matGC (*Figure 3E*), which reflects that these connections are not fully mature at this developmental stage, as previously shown (*Marín-Burgin et al., 2012*). Nonetheless, differences in inhibition between immature and mature neurons are greater than in excitation, giving higher mean excitation/inhibition ratios in 4wpiGC than matGC at 1 and 10 Hz (*Figure 3E*). At 20 Hz and 40 Hz higher ratios are only observed within the first pulses of the train, where spiking differences reside (*Figure 3D*, *Figure 3—figure supplement 1*). Therefore, higher excitation/inhibition ratios in 4wpiGC explain their higher levels of activation after train stimulation. On the other hand, in the absence of inhibition both 4wpiGC and mature GC display similar firing properties. This similarity results from the balance between a strong excitatory drive combined with a low input resistance in mature GC, and a weak excitation matched to high resistance in 4wpiGC. The latter combination facilitates reaching spiking threshold with small postsynaptic currents (*Mongiat et al., 2009*).

The different rising kinetics of the recruited inhibition between matGC and 4wpiGC determines different time windows in which inhibition restricts excitation. To better understand the extent to which this difference affects activation as a function of frequency, we calculated the residual inhibition at different periods corresponding to the studied frequencies, and added the recruited inhibition observed on average at 1 Hz on top (*Figure 3—figure supplement 2*). The results clearly show that due to the faster kinetics of the IPSC in matGC, the contribution of the recruited inhibition in matGC is higher than in 4wpiGC. Therefore, the sum of residual plus recruited inhibition in matGC is higher than excitation even if stimuli are 100 ms apart (restricting their responses to 10 Hz stimulation), whereas 4wpiGC have a wider temporal window in which inhibition does not overcome excitation (*Figure 3—figure supplement 2*).

Overall, these results show that the higher activation levels of 4wpiGC are due to a higher balance between the excitation and inhibition, which is mainly generated by a smaller inhibition at the time around spike generation.

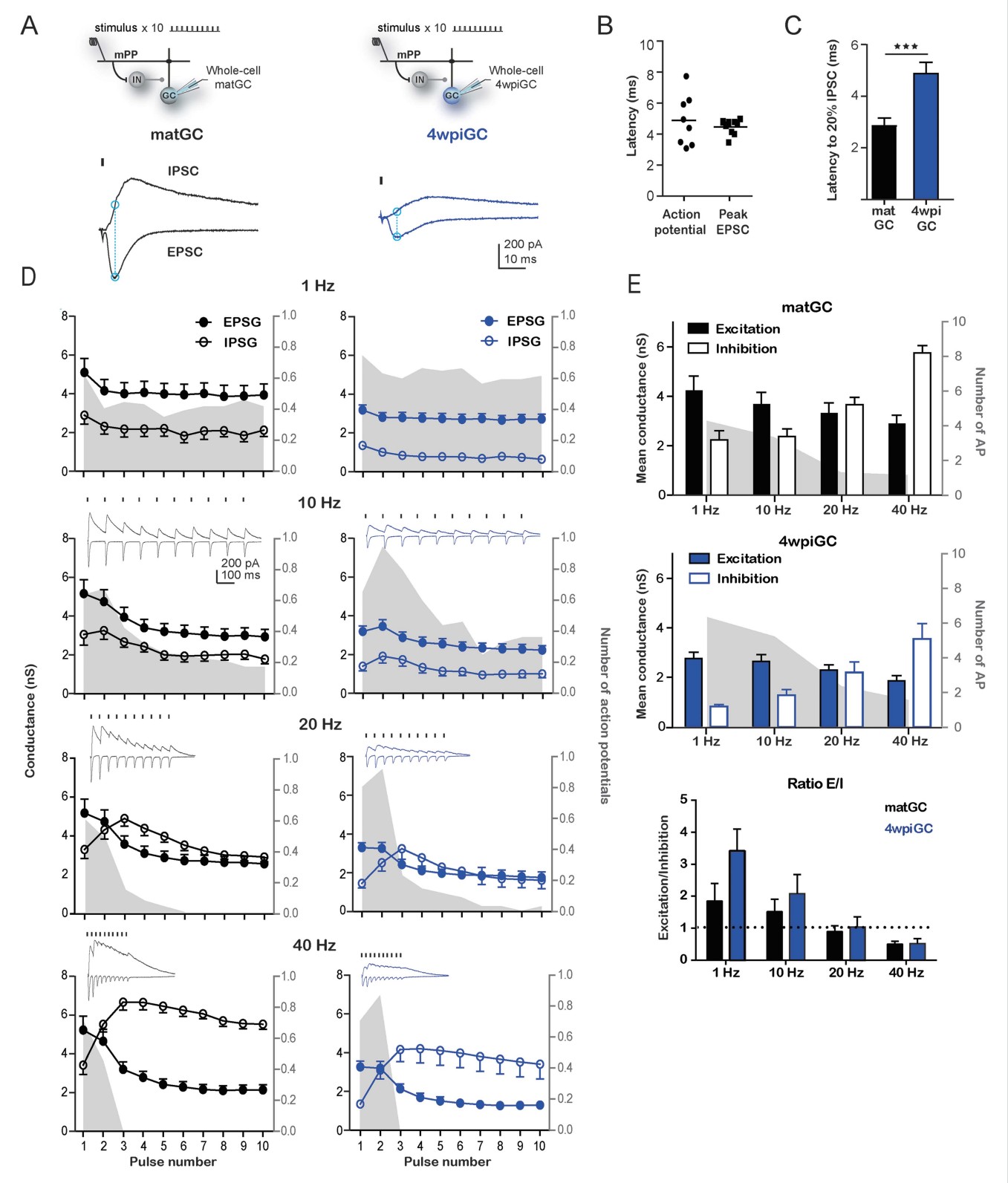

**Figure 3**. Interaction between excitation and inhibition determines the response of GC to frequency. (**A**) Top, the schemes show the recording configuration. Whole-cell voltage-clamp recordings were obtained from matGC and 4wiGC in response to stimulation of mPP with trains at different frequencies. Bottom, example of recordings. IPSCs were recorded at the reversal potential of excitation (−60 mV), and EPSCs were recorded at the reversal potential of inhibition (0 mV). The dash indicates the time of the stimulus. In subsequent quantifications, excitation and inhibition were measured

*Figure 3. continued on next page*

*Figure 3. Continued*

at the peak of the EPSC, marked with blue in the trace. Traces in black correspond to matGC and in blue to 4wpiGC. (**B**) Latency to action potentials compared to latency of the peak EPSC measured from the stimulating artifact. The action potentials and the peak of the EPSC occur close in time. (**C**) Average latency to reach the 20% of the peak IPSC evoked by the first stimulation pulse at 1 Hz in matGC (black) and 4wpiGC (blue). The IPSC is slower in 4wpiGC than in matGC (***$p < 0.001$, t test). (**D**) Average excitatory (EPSG) and inhibitory (IPSG) conductance evoked by each pulse in the train at 1 Hz, 10 Hz, 20 Hz, and 40 Hz. The values were calculated as indicated in **A**. For comparison, the spiking probability for each pulse in the train obtained from *Figure 1A* is plotted as the gray shadow in the back. The insets show representative current traces obtained for each frequency. Dashes indicate stimuli. (**E**) Average of the mean of excitatory and inhibitory conductances evoked by the 10 pulses in matGC and 4wpiGC. Excitation decreases with frequency and is higher in matGC than in 4wpiGC at all frequencies of stimulation (two-way ANOVA paired between frequencies; variation between GC: * $p < 0.05$; variation between frequencies. ***$p < 0.001$; interaction: ns, $p > 0.05$; N (4wpiGC) = 17 cells N (matGC) = 14 cells at the four frequencies). Inhibition increases with frequency and is higher in matGC than in 4wpiGC that at all frequencies of stimulation (two-way ANOVA paired between frequencies; variation between GC: ***$p < 0.001$; frequency variation. ***$p < 0.001$; interaction: ns, $p > 0.05$. N (4wpiGC) = 9 cells and N (matGC) = 11 cells at the four frequencies). For comparison, the average number of action potentials is plotted as a gray shadow in the back. Bottom: ratios between mean EPSG and IPSG evoked along train stimulation at 1, 10, 20, and 40 Hz for 4wpiGC (blue) and matGC (black). Dotted lines show the switch between higher excitation (above ratio = 1) and inhibition (below ratio = 1) balance. Error bars indicate SEM. Stimulation artifacts were erased from traces for better visualization.

The following figure supplements are available for figure 3:

**Figure supplement 1**. Excitation/inhibition ratios evoked by train stimulation.

**Figure supplement 2**. Contribution of residual and recruited inhibition in immature and mature GC.

## Inhibitory circuits involved in the activation of mature and immature GC

The above findings show that inhibitory circuits are responsible for generating differences in the responsiveness to different frequencies between mature and immature GC. Therefore, we focused on the understanding of which components of the activated inhibitory circuit are responsible for the observed differences. As we have mentioned, activation of mPP not only produces excitation on GC but also activates inhibitory circuits in a feedforward manner. In addition, activation of GC could also recruit feedback inhibition that could influence their activity, especially during trains of stimuli. The contribution of each type of inhibition could vary depending on the train frequency. To study the specific contribution of feedforward and feedback inhibition to the total inhibition recruited with trains of stimulation, whole-cell voltage-clamp experiments were performed in 4wpiGC and matGC, and trains were applied to the mPP at different frequencies. EPSC and IPSC were recorded in control conditions and in the presence of DCG4, an agonist of group II metabotropic glutamate receptors (mGluR$_{2/3}$) that reduces release probability in mossy fiber terminals (*Kamiya et al., 1996*). Application of DCG4 abolished feedback inhibition, and the remaining inhibition corresponded then to feedforward inhibition (IPSC-FF). The contribution of feedback inhibition (IPSC-FB) was then calculated by subtraction of the IPSC-FF to total IPSC (*Figure 4A*, *Figure 4—figure supplement 1*).

Quantification of the kinetics of the IPSC-FF and IPSC-FB recruited by the stimulation of the first pulse at 1 Hz reveals that the IPSC-FB is recruited after the IPSC-FF and after the peak of the EPSC (*Figure 4B*). Notably, the IPSC-FF is slower in 4wpiGC than in matGC. Conversely, the IPSC-FB does not show significant differences between 4wpiGC and matGC.

To compare IPSC-FF and IPSC-FB between matGC and 4wpiGC, we calculated the mean of the IPSG-FF and the IPSG–FB of the 10 pulses at the peak of the EPSC (*Figure 4C*). Interestingly, the results reveal differences between matGC and 4wpiGC only in the IPSG-FF, which is higher in the four studied frequencies of stimulation. However, the FB-IPSG does not differ between matGC and 4wpiGC at any frequency. These differences could be explained by the observed differences in the kinetics of FFI but not FBI.

Since DCG4 can also affect neurotransmitter release in the mPP pathway (*Macek et al., 1996*), and therefore, modify recruitment of inhibitory neurons in a feedforward manner, we tested the possibility that the recorded FF-IPSC in GC could be affected by DCG4. Parvalbumin (PV) interneurons are known to respond to the PP pathway and participate in feedforward inhibition (*Sambandan et al., 2010*). We studied the effect of DCG4 on the activation (spiking) of PV interneurons using hippocampal slices from PV-Tom mice (*Hippenmeyer et al., 2005*). As expected, PV interneurons already showed responses at very low stimulation intensities (10% fEPSP$_{slope}$). Thus, activation in

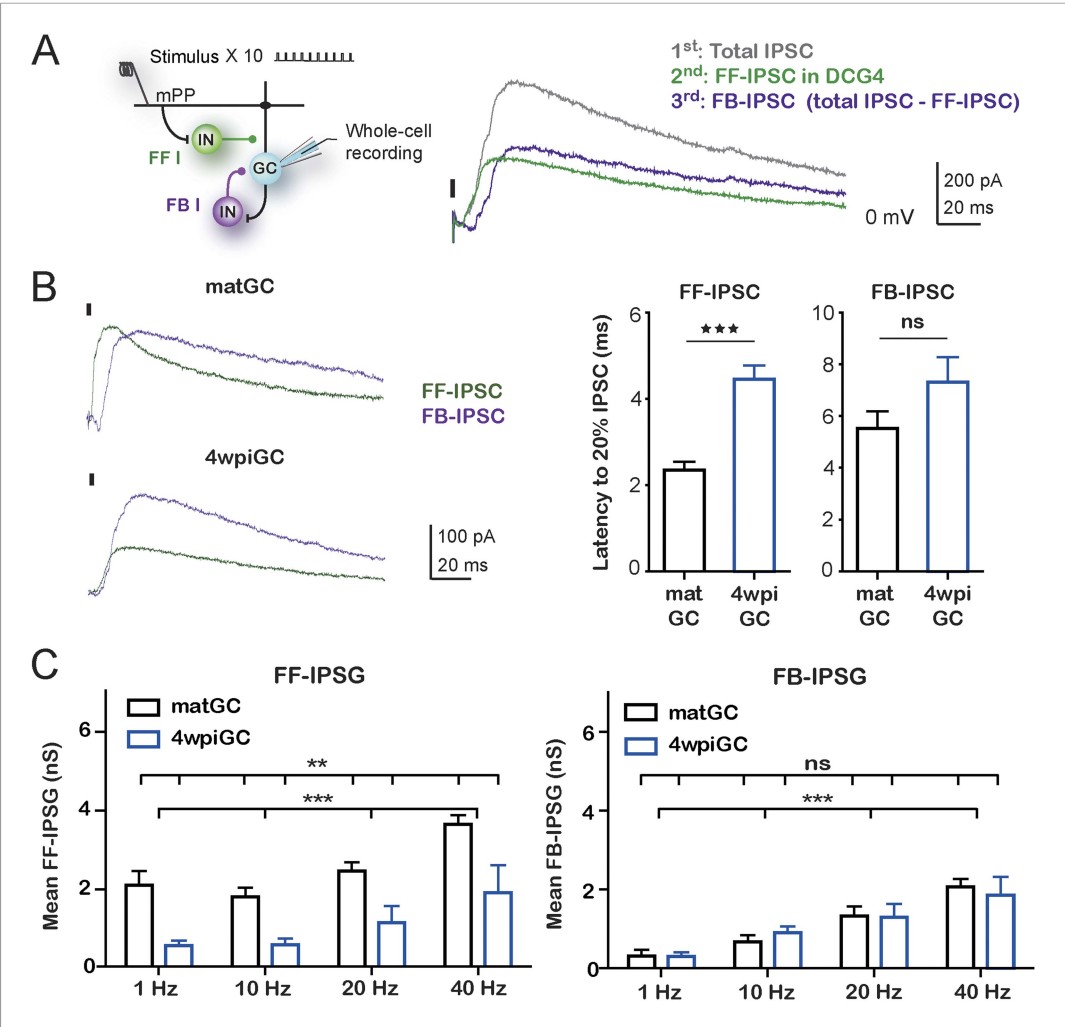

**Figure 4**. Feedforward inhibition generates differences between mature and immature GC. (**A**) Recording configuration showing the activated circuit. Stimulation of mPP-activated monosynaptic excitation and feedforward inhibition (FF I, green). Feedback inhibition (FB I, purple) is recruited by the activity of GC. Right traces show inhibitory currents recorded in whole cell, voltage clamping the cell at the reversal potential of excitation (~0 mV). The recorded total inhibitory post-synaptic current (total IPSC) obtained under control conditions is in gray. Application of DCG4 abolished feedback IPSC leaving only the feedforward IPSC (FF-IPSC, green trace). Feedback IPSC (FB-IPSC, purple trace) was measured by subtraction of the FF-IPSC from the total IPSC. (**B**) Left, traces from representative recordings of FF-IPSC and FB-IPSC from matGC and 4wpiGC. FF-IPSC in 4wpiGC is slower. The dash indicates the time of the stimulus. Right, average latency to reach 20% of the peak IPSC evoked by the first stimulation pulse at 1 Hz for FF and FB inhibition in matGC and 4wpiGC. The IPSC-FF is slower in 4wpiGC than in matGC (***$p < 0.001$, t test). The latency of the IPSC-FB did not show significant differences between 4wpiGC and matGC (ns, $p > 0.05$, t test). (**C**) Left, mean FF inhibitory postsynaptic conductance (IPSG-FF) evoked by stimulation of 10 pulses to mPP measured at the peak of the EPSC in matGC and 4wpiGC. The IPSG-FF increases with frequency and is higher in matGC than in 4wpiGC at all frequencies of stimulation (two-way ANOVA paired between frequencies; variation between GC: **$p < 0.01$; frequency variation. ***$p < 0.001$; interaction: ns, $p > 0.05$). Right, mean FB inhibitory postsynaptic conductance (IPSG-FB) evoked by stimulation of 10 pulses to mPP measured at the peak of the EPSC in matGC and 4wpiGC. The IPSG-FB shows no differences between matGC and 4wpiGC, but increases in both GC with frequency (two-way ANOVA paired between frequencies; variation between GC: ns, $p > 0.05$; variation between frequencies. ***$p < 0.001$; interaction: ns, $p > 0.05$). N (4wpiGC) = 6–7 cells, N (matGC) = 10–11 cells, in the four frequencies. Error bars indicate SEM. Stimulation artifacts were erased from traces for better visualization.

The following figure supplements are available for figure 4:

*Figure 4. continued on next page*

*Figure 4. Continued*

**Figure supplement 1**. Feedforward and feedback inhibition recruited by train stimulation.

**Figure supplement 2**. Response of PV+ interneurons to mPP stimulation is not affected with DCG4.

response to stimulation of the mPP at the intensities used in the experiments (50% fEPSP$_{slope}$) was well above threshold and unaltered by the application of DCG4 (*Figure 4—figure supplement 2*).

The above results indicate that the differences in the recruitment of mature and immature neurons are dictated specifically by feedforward inhibition and are given by the fact that the IPSC-FF is slower in 4wpiGC compared to matGC. However, both IPSG-FB and IPSG-FF increase with frequency, indicating that both types of inhibition contribute to generate differences in activation among frequencies.

## Spiking in immature GC is less time locked to the stimulus

Immature neurons are better followers of spike trains arriving to the mPP due to a weak influence of inhibition. However, a weak and slower inhibition can lead to increased temporal variation in the responses, since temporal fidelity is ensured by inhibition (*Pouille and Scanziani, 2001*). The slower inhibition in immature neurons would allow a wider temporal window in which neurons can summate excitation to spike. Therefore, we compared the latency and the jitter of action potentials evoked in matGC and 4wpiGC in response to the first pulse of each stimulus (*Figure 5A*).

Indeed, mature neurons show a time locked response to afferent stimulation, since the majority of mature cells is activated after 3 ms of the stimulation of the mPP and all neurons tend to spike around the time of the population spike (pop spike) (*Figure 5B*). Immature neurons, on the contrary, show longer latencies (most cells spike after the pop spike) and broader distribution of latencies to spike than mature neurons. In addition, 4wpiGC also show a higher variance of their spike times than mature neurons. The little jitter of matGC indicates that this population of neurons has a higher temporal fidelity (*Figure 5C*). In the absence of inhibition, both populations increased their spike time variability, indicating that inhibition restricts their temporal responses (*Figure 5D*). Even in the absence of inhibition mature neurons show more precise activation which could be given by the nature of the excitatory input, which is stronger in mature GC than in immature GC. On the other hand, immature neurons have slower membrane time constant (*Mongiat et al., 2009*) that could also contribute to their higher spiking jitter. These results suggest a division of labor in the populations of immature and mature GC, in which immature neurons respond with higher efficacy to the frequency of the afferent inputs but with imprecise responses to the timing of the afferent spike, while mature neurons respond with a narrow tuning to the time of the afferent inputs, though with less efficiency to respond to frequency.

## Discussion

In this work, we reveal that mPP stimulation with trains of stimulus at physiologically relevant frequencies evoke higher levels of activation in immature adult born hippocampal neurons, such that they are more reliable in transmitting different frequencies. The results also demonstrate that the activation differences between immature and mature GC are governed by the recruitment of feedforward inhibitory circuits. The small and slow inhibition, that allows immature neurons to be good transmitters of frequency, together with the weak excitation they receive, generates a worse temporal fidelity in those cells compared to mature neurons, that are time locked to the stimulus. Thus, activity arriving to the hippocampus at different frequencies could be reflected by the activation of graded populations of neurons, immature GC that are reliable transmitters of the incoming frequency, and mature GC that are precise at informing the beginning of the stimulus, but with sparse activity.

### Frequency-dependent activity of neurons

Numerous studies have shown that brain activity occurs with certain temporal patterns and proposed that the timing of the signals is itself a code (*Ahmed and Mehta, 2009*; *Singer, 2009*). Extracellular field potentials recorded in vivo present oscillations that change with different brain states

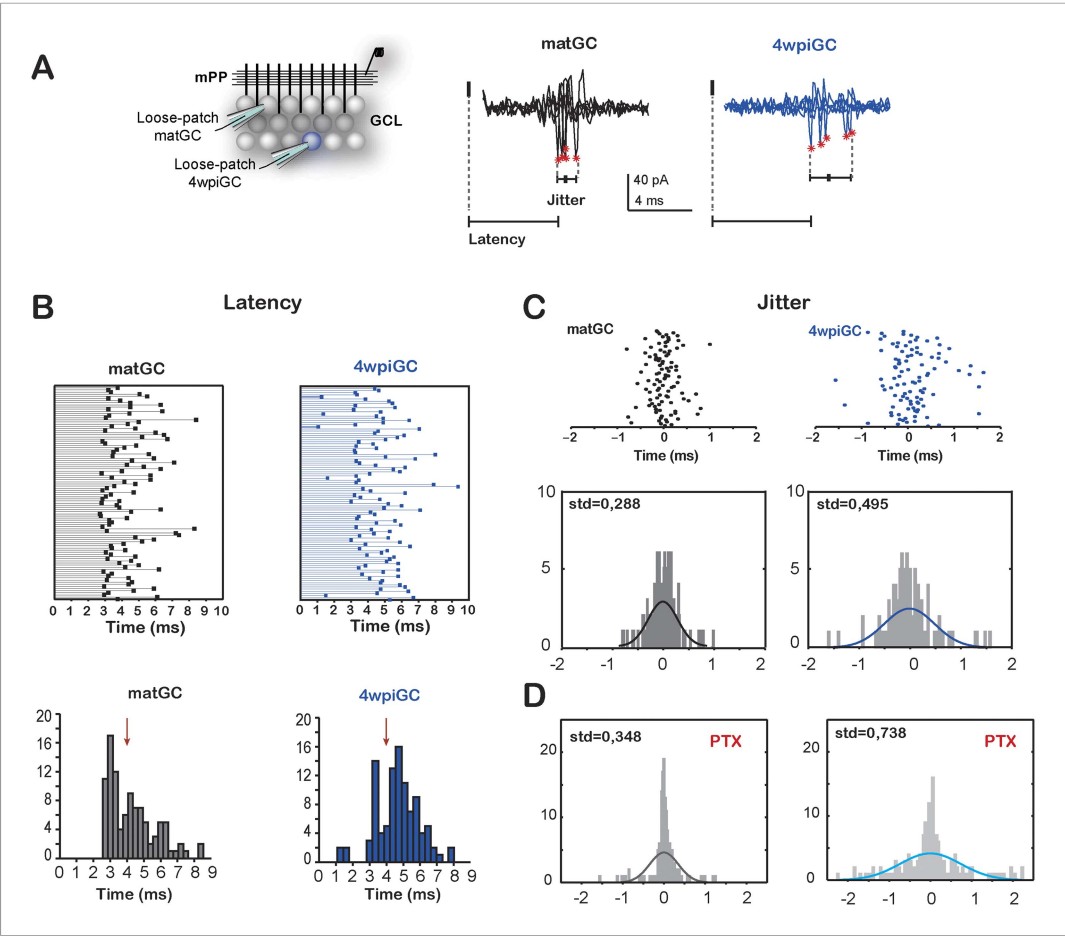

**Figure 5.** Temporal fidelity in mature and immature GC (**A**) Recording configuration. Left, the mPP was stimulated at an intensity of 50% fEPSP. Recordings were performed from 4wpiGC and matGC in loose patch configuration. Right, representative traces of evoked action potentials in response to stimulation of the mPP. The latency from the stimulation artifact to action potentials (asterisks) and spiking jitter and were calculated for matGC (black) and 4wpiGC (blue). (**B**) Top, latencies measured in 100 spikes from matGC (black) and 4wpiGC (blue). Bottom, distribution of latencies binned along 20 bins. The red arrow denotes the average pop spike time of all included experiments (4.36 ± 0.19 ms). (**C**) Top, jitter measured as the time of action potential occurrence relative to the mean spike timing of 3–5 trials. Bottom, jitter distribution in matGC and 4wpiGC; solid lines represent a normal fitting lines. Distribution is wider in 4wpiGC than in matGC (p < 0.001, two-sample F-test for equal variances). N (matGC) = 114 spikes from 8 cells; N (4wpiGC) = 96 spikes from 6 cells. Std: standard deviation. (**D**) Jitter distribution in matGC and 4wpiGC in the presence of picrotoxin (PTX); solid lines represent normal fitting lines. Distribution is wider in 4wpiGC than in matGC (p < 0.001, two-sample F-test for equal variances) and is wider in the presence of PTX for both matGC (p < 0.05, two-sample F-test for equal variances) and 4wpiGC (p < 0.001, two-sample F-test for equal variances). N (matGC) = 142 spikes of 9 cells; N (4wpiGC) = 170 spikes from 10 cells.

(*Womelsdorf et al., 2014*). For example, layer II of the EC and the DG of the hippocampus, tend to oscillate in frequencies theta (~7 Hz) and gamma (~40 Hz) (*Roux and Buzsaki, 2015*). Neurons from EC, the main afferents to DG, are activated by brief bursts of frequencies in the gamma order spaced by intervals in the order of theta frequency (*Chrobak and Buzsaki, 1998*; *Fyhn et al., 2004*; *Hafting et al., 2005*; *Bonnevie et al., 2013*). Our results indicate that differences in the activation frequency of afferent MEC inputs arriving to DG differentially activate immature and mature GC. Although both populations of neurons tend to behave as low pass filters, immature neurons are more efficient in responding to trains (*Figure 1*). In fact, immature neurons would be able to spike with more than two spikes in a theta cycle, whereas mature neurons would not. The implications of this result are important also in terms of the effect that GC could have in CA3. It has been shown that direct

stimulation of one GC is capable of activating a pyramidal cell in CA3; however, one action potential is not enough to cause activation of the pyramidal cell, but it requires at least two, and the activation probability increases exponentially with the frequency of GC stimulation (*Henze et al., 2002*). Because the connectivity of 4wpiGC in CA3 is very similar to the connectivity of matGC (*Temprana et al., 2015*), our results suggest that immature GC could be more efficient in transmitting high frequency stimuli which can cause a robust activation of CA3 pyramidal cells. A higher probability of efferent activation by immature GC may have a role to reinforce these connections, since 4wpiGC also have greater levels of LTP in these synapses (*Gu et al., 2012*), suggesting that perhaps immature neurons could select networks in CA3 that correspond to a particular moment in which those GC are still immature.

## Inhibitory circuits in the control of activity

Local inhibitory circuits are primarily responsible for differences in activation generated by trains of stimuli. The large increase in inhibition at high frequencies determines very low levels of activation. Thus, due to inhibitory circuits, the activity arriving to DG at different frequencies can be differentially decoded by mature and immature GC.

We observed that at low frequencies, feedforward inhibition plays an important role in controlling activation whereas feedback inhibition is very low. As frequency increases, feedback inhibition also increases and adds to the control of activity at high frequencies.

Despite the difference in intrinsic properties, activation differences between immature and mature GC are entirely generated by inhibition. The relation between recruited excitation and inhibition is crucial to determine GC activation (*Marín-Burgin et al., 2012*; *Dieni et al., 2013*). Therefore, the slow feedforward inhibition observed in immature neurons is essential to determine differences in activation between them and mature neurons. The time interval between trains of pulses at 1 and 10 Hz stimulation is such that the delayed IPSC in 4wpiGC does not compensate the EPSC, and thus inhibition at these frequencies does not affect the activation of immature cells. This could have important functional consequences in light of studies suggesting that activation of DG is set to frequencies near theta (*Jung and McNaughton, 1993*; *Leutgeb et al., 2007*; *Neunuebel and Knierim, 2012*). At frequencies higher than 10 Hz, the time interval between pulses is such that the IPSC recruited pulse to pulse can summate and exceedingly compensate the EPSC in both mature and immature GC.

Additionally, the growth of inhibitory currents in GC at high frequencies could be the result of specific recruitment of different types of interneurons affecting equally both GC. A recently published study in mature GC shows that dendritic inhibition is preferentially recruited with high frequency stimulation and grows along the train of pulses. Instead perisomatic inhibition is recruited at low and high frequencies, and decreases within a train (*Liu et al., 2014*). In the same line, we have previously shown that dendritic inhibition is very similar between mature and immature neurons, but perisomatic inhibition is bigger and faster in mature GC (*Marín-Burgin et al., 2012*). In light of these results and since we have found that the main difference between mature and immature GC resides in feedforward inhibition, it is highly possible that feedforward inhibition has an important perisomatic component, like the one PV interneurons exert. The slow inhibition on 4wpiGC could be a consequence of the immature nature of perisomatic inhibitory contacts from PV interneurons, which would require future investigation. Also, there could be a differential contribution of mossy cells, which can affect inhibition and excitation on GC (*Chancey et al., 2014*).

Our results show, in addition, that spikes in mature neurons are more time-locked to the incoming input than immature neurons, partly due to inhibition, but also probably to the stronger excitatory synapses they receive, as has been suggested. In vivo experiments, recording in the DG in animals with reduced neurogenesis show that they have reduced number of neurons with spiking broadly tuned to theta cycle, and most cells show a narrow tuning to a specific phase of theta, suggesting that immature cells could presumably contribute to the broadly tuned population. This notion may have functional implications in the influence that mature and immature neurons could have on CA3 (*Rangel et al., 2013*).

It has been proposed that there are two coding strategies in the nervous system (*Harris and Mrsic-Flogel, 2013*), a *sparse code*, in which information is encoded at any instant by the spiking of a small subset of neurons within the population, in general with low mean firing rates and high selectivity, that

could increase information storage. And a *dense code*, in which most neurons are active at any moment and information is encoded by variations in firing rate. Our results suggest that mature and immature GC could represent these two different codes coexisting in the same structure and presumably serving at different functions. While mature GC could specifically represent a particular input, immature GC, with a higher ability to drive CA3, could be generating networks in CA3 that would presumably represent a particular experience occurring while they are immature.

## Materials and methods

### Viral vectors preparation

A replication-deficient retroviral vector based on the Moloney murine leukemia virus was used to express RFP or GFP under a CAG promoter (*Marín-Burgin et al., 2012*). Retroviral particles were assembled using three separate plasmids containing the capside (CMV-vsvg), viral proteins (CMV-gag/pol), and the transgene (CAG-RFP or CAG-GFP). Plasmids were transfected onto HEK 293T cells using deacylated polyethylenimine. Virus-containing supernatant was harvested 48 hr after transfection and concentrated by two rounds of ultracentrifugation. Virus titer was typically $\sim10^5$ particles/µl.

### Animals and surgery for retroviral delivery

Female C57Bl/6J mice 6–7 weeks of age were housed at 4 mice per cage, with two running wheels. Running wheel housing started 2–4 days before surgery and continued until the day of slice preparation. For surgery, mice were anesthetized (150 µg ketamine/15 µg xylazine in 10 µl saline/g), and virus (1 µl at 0.125 µl/min) was infused into the dorsal area of the right DG using sterile microcapillary calibrated pipettes and stereotaxic references (coordinates from bregma: −2 mmanteroposterior,-1.5 mm lateral, −1.9 mm ventral). Animals were killed for acute slice preparation 4 weeks after the surgery.

To generate $Pvalb^{Cre}$;CAG$^{FloxStopTom}$ (PV-Tom) mice, *B6;129P2-Pvalbtm1(cre)Arbr/J (PV$^{cre}$)* mice (*Hippenmeyer et al., 2005*) were crossed to *B6.Cg-Gt(ROSA)26Sortm14(CAG-tdTomato)Hze/J (Ai14)* conditional reporter mice. Animals heterozygous for Cre and tomato were used.

Experimental protocols were approved by the Institutional Animal Care and Use Committee Leloir Foundation (Protocols Number 2009 08 37 and 64/2015, IACUC, Leloir Institute Foundation) according to the Principles for Biomedical Research involving animals of the Council for International Organizations for Medical Sciences and provisions stated in the Guide for the Care and Use of Laboratory Animals.

### Slice preparation

Mice were anesthetized and decapitated at 4 weeks post injection (wpi). Brains were removed into a chilled solution containing (mM) 110 choline-Cl⁻, 2.5 KCl, 2.0 $NaH_2PO_4$, 25 $NaHCO_3$, 0.5 $CaCl_2$, 7 $MgCl_2$, 20 dextrose, 1.3 Na⁺-ascorbate, 3.1 Na⁺-pyruvate, and 4 kynurenic acid. The right hippocampus was dissected and slices (400 µm thick) were cut transversally to the longitudinal axis in a vibratome and transferred to a chamber containing artificial cerebrospinal fluid (ACSF; mM): 125 NaCl, 2.5 KCl, 2.3 $NaH_2PO_4$, 25 $NaHCO_3$, 2 $CaCl_2$, 1.3 $MgCl_2$, 1.3 Na⁺-ascorbate, 3.1 Na⁺-pyruvate, and 10 dextrose (315 mOsm). Slices were bubbled with 95% $O_2$/5% $CO_2$ and maintained at 30°C for >1 hr before experiments started. Salts were acquired from Sigma-Aldrich (St. Louis, MO).

### Electrophysiological recordings

Recorded neurons were visually identified by fluorescence and infrared DIC videomicroscopy. The mature neuronal population encompassed RFP⁻ neurons localized in the outer third of the GCL (*Mongiat et al., 2009*). Whole-cell recordings were performed using microelectrodes (4–5 MΩ) filled with (in mM) 130 CsOH, 130 D-gluconic acid, 2 $MgCl_2$, 0.2 EGTA, 5 NaCl, 10 HEPES, 4 ATP-tris, 0.3 GTP-tris, 10 phosphocreatine. In experiments where the intrinsic responses to current pulses were evaluated, a potassium gluconate internal solution was used (in mM): 120 potassium gluconate, 4 $MgCl_2$, 10 HEPES buffer, 0.1 EGTA, , 5 NaCl, 20 KCl, 4 ATP-tris, 0.3 GTP-tris, and 10 phosphocreatine (pH = 7.3; 290 mOsm). Loose-patch recordings were performed with ACSF-filled patch pipettes (5–6 MΩ). Field recordings were performed using patch pipettes (5 MΩ) filled with 3 M NaCl. Recordings were obtained using Axopatch 200B and Multiclamp 700B amplifiers, (Molecular Devices, Sunnyvale, CA), digitized, and

acquired at 20 KHz onto a personal computer using the pClamp10 software. Membrane capacitance and input resistance were obtained from current traces evoked by a hyperpolarizing step of 10 mV. Series resistance was typically 10–20 MΩ, and experiments were discarded if higher than 30 MΩ.

## Calibration of the input strength for the mPP stimulation

The input strength is proportional to the number of activated mPP axons, and it was assessed as percentage of the field excitatory postsynaptic potential slope (fEPSP$_{slope}$, *Figure 1—figure supplement 1*), which increases linearly with the fiber volley (*Figure 1—figure supplement 1B*, inset). Unlike the fiber volley, the fEPSP can be well visualized in all recordings. For this purpose, a field-recording microelectrode was placed on the granule cell layer (GCL) to record the fEPSP and the pop spike in response to the mPP stimulation. To compare input strengths across experiments, the fEPSP$_{slope}$ elicited at any given stimulus intensity was normalized to the fEPSP$_{slope}$ evoked at a stimulus intensity that evokes a maximal pop spike (100%, *Figure 1—figure supplement 1A,B*). Input strength was kept at 50% for all experiments and frequencies (*Figure 1—figure supplement 1C*). A similar approach for input strength calibration has been previously used (*Pouille et al., 2009*; *Marín-Burgin et al., 2012*). The input strength was always assessed and calibrated in the absence of PTX. Thus, for a given input strength within a slice, the number of mPP axons stimulated in control conditions or in the presence of PTX (100 µM, Sigma-Aldrich) were the same.

## Evoked postsynaptic currents and conductances

Evoked monosynaptic EPSCs and disynaptic IPSCs were recorded after mPP stimulation at 50% input strength. To minimize the contribution of IPSCs mediated by direct stimulation of inhibitory axons, we only considered experiments in which kynurenic acid (KYN) (6 mM, bath applied at the end of the experiment) blocked >70% of IPSCs. EPSCs were isolated by voltage clamping GC at the reversal potential of the IPSC measured for each individual neuron (~60 mV). In turn, disynaptic IPSCs were recorded at the reversal potential of the EPSC (~0 mV). When present, direct monosynaptic IPSC recorded in KYN (always <30% of the peak IPSC amplitude) was subtracted from the IPSC. Synaptic excitatory and inhibitory conductances were computed as the EPSC or IPSC divided by the driving force at which the synaptic currents were recorded. 2-amino-5-phosphonovaleric acid (AP5, 50 mM, Tocris, United Kingdom) was perfused during recordings to block NMDA currents. We ruled out a possible influence of NMDA receptor-mediated currents on the activation of GC by recording spiking in the presence of AP5. No differences were found between spiking in ACSF and after application of AP5 in the bath (data not shown).

## Calculation of efficacy of frequency transmission

Efficacy was computed as the average of the probability of occurrence of action potentials in the same frequency range as that given by the stimulus. Thus, a granule cell that responds with five consecutive action potentials in response to the train of 10 pulses, has an efficacy = 4/9 (4 intervals were represented out of the 9 that were in the train), while a GC with five discontinuous action potential firing, (without presenting two continuous) has an efficacy of 0/9. Efficacy was calculated for matGC and 4wpiGC (*Figure 1—figure supplement 2*).

## Measurement of feedforward and feedback inhibition

To dissect feedforward and feedback inhibitory components, after registering the total inhibition evoked by the stimulation of the mPP, mossy fiber synapses were blocked with DCG4 (1 µM, Tocris) (*Kamiya et al., 1996*). In this condition, the remaining feedforward current was measured, and the feedback component was calculated by subtracting the feedforward current to the total current offline. In parallel experiments, we controlled for the effect of DCG4 on the activation of inhibitory interneurons, by measuring the change in their activation in response to mPP stimulation after the application of DCG4, using a stimulation intensity of 50% fEPSP as in the rest of the work (*Figure 4—figure supplement 2*). To record from identified interneurons, we used transgenic mice with fluorescently labeled PV interneurons, PV-Tom mice.

## Statistical analysis

Significant differences were assessed by two-tailed *t*-test, Wilcoxon signed-rank test, two-sample F-test for equal variances, one-way or two-way ANOVA, and Bonferroni's post-hoc test, as indicated in the figure legends.

## Acknowledgements

We thank Guillermo Lanuza for B6;129P2-Pvalbtm1(cre)Arbr/J (PV*cre*) and B6.Cg-Gt(ROSA)26Sortm14 (CAG-tdTomato)Hze/J (Ai14) mice; Natalia Beltramone for viral preparations; and Emilio Kropff, Sung Min Yang, and Guillermo Lanuza for insightful discussions as well as all members from AFS and AMB labs. AMB and AFS are investigators of the National Research Council (CONICET). MBP and MBO were supported by CONICET fellowships. This work was supported by grants from Argentine Agency from the Promotion of Science and Technology (PICT 2010-1110 and PICT 2013-0182 to AMB; PICT 2010-1978 to AFS), Structural Convergence Fund for MERCOSUR (FOCEM), the National Institutes of Health (FIRCA R03TW008607-01 to AFS), and the Howard Hughes Medical Institute (SIRS grant#55007652 to AFS).

## Additional information

### Funding

| Funder | Grant reference | Author |
|---|---|---|
| Ministerio de Ciencia, Tecnología e Innovación Productiva | PICT 2010-1110 and PICT 2013-0182 | Antonia Marin-Burgin |
| Structural Convergence Fund for MERCOSUR | FOCEM | Antonia Marin-Burgin |
| National Institutes of Health (NIH) | FIRCA (R03TW008607-01) | Alejandro F Schinder |
| Howard Hughes Medical Institute (HHMI) | SIRS grant#55007652 | Alejandro F Schinder |
| Ministerio de Ciencia, Tecnología e Innovación Productiva | PICT 2010-1978 | Alejandro F Schinder |

The funders had no role in study design, data collection and interpretation, or the decision to submit the work for publication.

### Author contributions

MBP, MBO, Conception and design, Acquisition of data, Analysis and interpretation of data, Drafting or revising the article; AFS, AM-B, Conception and design, Analysis and interpretation of data, Drafting or revising the article

### Ethics

Animal experimentation: Experimental protocols were approved by the Institutional Animal Care and Use Committee of the Fundación Instituto Leloir (Protocols Number 2009 08 37 and 64/2015, IACUC, Leloir Institute Foundation) according to the Principles for Biomedical Research involving animals of the Council for International Organizations for Medical Sciences and provisions stated in the Guide for the Care and Use of Laboratory Animals.

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
