## [Decision Letter]

Thank you for submitting your work entitled "Differential inhibition onto developing and mature granule cells generates high-frequency filters with variable gain" for review at *eLife*. Your article has been favorably evaluated by a Senior editor, and three reviewers, one of whom is a member of our Board of Reviewing Editors.

Your manuscript was carefully read by three reviewers who were very positive about the new finding on the differential frequency-dependent recruitment of young and mature GCs by afferent perforant path inputs. However some major criticisms have been formulated which I would like to summarize in the following.

1) The reviewers were wondering how much the observed frequency-dependent recruitment of GCs depends on the intrinsic properties of young vs. mature cells. This aspect should be addressed in the Discussion section.

2) The reviewers ask why inhibition fails to influence the activity and the definition of the temporal window for action potential generation in young cells. The reviewers would have expected that although weak, it should have some effect on the activity of the young cells. Is inhibition indeed as weak as shown in the study or is the applied protocol insufficient to observe functional consequences?

3) Concerns related to the normalization paradigms were raised and should be addressed by the authors in the revised manuscript.

4) The relationship between excitation and inhibition seems to be similar between both young and mature GCs. A better presentation of the interaction between excitation and inhibition in Figure 3 is required and could be provided by excitation/inhibition ratios.

5) The data related to the effects of DCGIV are interesting, however, the authors should discuss the mild but existing (and published) blocking effects of DCGIV on perforant path-mediated inputs onto GCs.

6) Since there seems to be some overlap in the results between previous papers from the same and other groups, the novel aspects of the study should be emphasized and better discussed in the revised manuscript.

6) The influence of running wheels on the obtained data remained unclear and should be addressed in the Discussion. Please see below the original major criticisms of the reviewers:

*Reviewer 1*:

1) The authors argue that young and mature GCs have both the intrinsic ability to discharge at high frequency on the basis of square current injections in the soma. However, this is a different situation than stimulating PP inputs. Young GCs have higher input resistances, slower membrane time constants than mature ones. Thus, their responsiveness to input frequency should be different. Indeed, young GCs have been demonstrated to be more responsive/excitable than mature ones due to lower activation threshold by the authors in Marin-Burgin, Science 2012. Please revise your statement or alternatively discuss the influence of intrinsic properties on the results.

2) In the subsection “Inhibitory control of activity in mature and immature GC”: please mention that the broader window of excitation of young GCs is not only dependent on the weaker inhibition but also on the intrinsic properties such as slower membrane time constant of young GCs.

3) It remains a bit unclear why the authors differentiate between residual and recruited inhibition because at the end they sum both and compare them with synaptic excitation. The authors argue that a differentiation between residual and induced inhibition needs to be performed because of different multiple-pulse dynamics of synaptic inhibition. However, the authors do not provide information on the extent of these differences and how they may potentially impact total inhibition and thus discharge activity.

4) Which IN types may contribute to the feedforward inhibition? The authors argue that PV-INs are recruited by PP stimulation. However, the time course of inhibition is slower than for feedback inhibition. This stays in contrast to the perisomatic location of PV-IN synapses. Therefore, the authors should discuss more precisely which IN types could potentially contribute to FF inhibition in this circuitry.

*Reviewer 2*:

1) The authors conclude that immature GCs have a wider time window in which excitation can elicit spikes without being affected by inhibition (subsection “Inhibitory control of activity in mature and immature GC”) but it is not clear why the inhibitory conductance fails to provide functional inhibition. Either there is interesting biology that renders immature GCs insensitive to inhibitory conductances, or there is a less interesting (and potentially misleading) possibility that the stimulus paradigm is insufficient to detect the functional consequence of the inhibitory conductance. Even the relatively small conductance shown in Figure 3 in 4wpi cells should provide some degree of inhibition at 1 and 10 Hz.

2) In light of the above concern, the authors should carefully consider the use of their stimulus normalization paradigm. This procedure was apparently developed for CA1, where most cells can be driven to spike whereas in the DG this appears not to be the case (the authors suggest that only ∼ 50% of cells spike with inhibition intact). Results from this group's previous study (Supplementary Figure S2E Marin-Burgin; 2012) shows the non-linear relationship between stimulus intensity and % of spiking cells, so it is hard to understand how the pSpike (i.e., the % of spiking cells) provides a sensitive measure of the number of active perforant path fibers. The normalization paradigm shown in Figure 1—figure supplement 1 also does not show that the pSpike is maximal (panel B), and there is a very poor fit of the fEPSP data (panel C). It is not clear why the normalization procedure is required for these experiments, but if it is, the authors should confirm that the fEPSP is linearly related to number of active PP fibers (that could be assayed by the fiber volley) and that their measure of input strength is unaffected by blockade of inhibition. It is important to have similar stimulation parameters to compare the effect of GABAergic inhibition shown in Figure 2 with the results of Figure 1. An alternative approach would be to compare the effect of PTX on spiking in each cell to allow a within-cell comparison, which would obviate the need for the normalization paradigm.

3) The main conclusion is that inhibition dictates the frequency-dependent firing characteristics of mature and 4 wpi cells. But this is not immediately obvious from the data in Figure 3, since the frequency-dependent patterns of excitation and inhibition shown in 3D and E appear to be very similar in mature and 4wpi GCs (despite their different spiking patterns). This could be an issue of presentation – perhaps illustrating frequency-dependent differences in the E/I ratio between mature and 4 wpi cells within the same graph would make this point more clear.

In addition, it appears that the synaptic currents in Figure 3 were recorded in AP5 but the spiking data was not. Since the NMDAR component could become a source of depolarization during high frequency stimulation, it should be included in the synaptic analysis or the authors should rule it out by demonstrating a lack of effect of AP5 on spiking.

4) The dissection of feed-forward and feedback components of inhibition is potentially very interesting, but I am not convinced by the method since it relies entirely on the selective blockade of mossy fiber synapses by DCG-IV. In fact, 1 micromolar DCG-IV is well known to block medial peforant path evoked EPSCs (i.e. [30] and many subsequent studies). The authors should repeat this critical control experiment using EPSCs rather than spiking, since supra-threshold excitation could potentially mask a suppression of MPP synaptic strength that would confound all the analysis shown in Figure 4. Furthermore, since GC spiking presumably is required for feedback inhibition, it is difficult to understand how the feedback IPSG is greatest during 40 Hz stimulation, which is the frequency when GCs spike the fewest action potentials (compare Figure 1 with Figure 4). Is there a frequency-dependent recruitment of feedback interneurons? Showing the FB-IPSC during the train might be useful in this regard.

*Reviewer 3*:

1) It is not clear how the use of running wheels in all the experiments affects the interpretation of the data. There is growing evidence that suggests that running changes rates of maturation of DGCs and afferent connectivity (Snyder 2009, Piatti 2011, Bergami 2015). Since one of the aims of this study is to probe differences between immature adult-born DGCs and mature DGCs, it is important that the authors directly address whether running has an effect on activation of 4 weeks old adult-born DGCs following mPP stimulation at different frequency-trains (by comparing with a non-running group).

2) The notion that intrinsic properties, but local circuitry underlying inhibition and excitations, determines recruitment of immature and mature DGCs in response to PP stimulation is not new and has been addressed previously in Dieni et al., JNSc 2013. In this study, the authors used a dual pathway stimulation protocol (MML and OML) and found that ratio of inhibitory and excitatory drive onto DGCs is influenced by stimulus frequency and dictates spiking probability. The relevance of the main findings of this study should be integrated into the discussion of results here.

3) The extent of mossy cell contribution to activation of DGCs following mPP stimulation in slices is not clear. Does stimulation of EC Layer II inputs to DG differentially activate 4 week vs. mature DGCs at 20 Hz? This caveat should be discussed.

---

## [Author Response]

Reviewer 1:

*1) The authors argue that young and mature GCs have both the intrinsic ability to discharge at high frequency on the basis of square current injections in the soma. However, this is a different situation than stimulating PP inputs. Young GCs have higher input resistances, slower membrane time constants than mature ones. Thus, their responsiveness to input frequency should be different. Indeed, young GCs have been demonstrated to be more responsive/excitable than mature ones due to lower activation threshold by the authors in*
*Marin-Burgin, Science 2012**. Please revise your statement or alternatively discuss the influence of intrinsic properties on the results*.

The reviewer is correct in regard to the higher input resistance of immature GC. A direct consequence of this property is that they are more efficient at translating synaptic currents into membrane depolarization, even though they receive a weaker excitatory drive (due to the reduced number of contacts) as shown in [33]. Therefore, in the absence of inhibition they display similar spiking probability as mature neurons in response to excitatory inputs, as we show in Figure 2. Thus, despite having different intrinsic properties, their different responses to synaptic inputs are primarily mediated by inhibition.

We added some clarifications in the text to further emphasize this idea in the Results and Discussion sections.

*2) In the subsection “Inhibitory control of activity in mature and immature GC”: please mention that the broader window of excitation of young GCs is not only dependent on the weaker inhibition but also on the intrinsic properties such as slower membrane time constant of young GCs*.

As we stated in the previous point, the main difference in activation between young and mature GC is due to the inhibitory network rather than to intrinsic properties (as shown in Figure 2). However, differences in intrinsic properties can be better visualized in the data shown in Figure 5. Although immature neurons reach similar spiking levels of that of mature cells (Figure 2), spiking jitter is broader (Figure 5) which could be given by the slower membrane time constant of young cells. We added that concept at the end of Results: “… immature neurons have slower membrane time constant (33) that could also contribute to their higher spiking jitter”.

*3) It remains a bit unclear why the authors differentiate between residual and recruited inhibition because at the end they sum both and compare them with synaptic excitation. The authors argue that a differentiation between residual and induced inhibition needs to be performed because of different multiple-pulse dynamics of synaptic inhibition. However, the authors do not provide information on the extent of these differences and how they may potentially impact total inhibition and thus discharge activity*.

We thank the reviewer for the observation. We agree that it was not very clear on the text the necessity to differentiate between recruited and residual inhibition. We changed the subsection “Synaptic mechanisms underlying frequency dependence and responsiveness of immature and mature GC“ to better explain how the different rising kinetics of the recruited inhibition between matGC and 4wpiGC determines a different time window in which inhibition restricts excitation.

*4) Which IN types may contribute to the feedforward inhibition? The authors argue that PV-INs are recruited by PP stimulation. However, the time course of inhibition is slower than for feedback inhibition. This stays in contrast to the perisomatic location of PV-IN synapses. Therefore, the authors should discuss more precisely which IN types could potentially contribute to FF inhibition in this circuitry*.

We think that the slow FF inhibition we observe in 4wpiGC corresponds to perisomatic slower IPSC because in a previous study we directly stimulated perisomatic inhibition and found postsynaptic currents with slower kinetics in immature GC (Marin Burgin et al. 2012). The synaptic mechanism underlying the slower inhibition in 4wpiGC would require further investigation, but we hypothesize that perisomatic inhibition is incipient at this neuronal age and could, thus, be immature and physiologically different. We think that differences could still reside on PV+ interneuron contacts. However, we would need to investigate in detail that synapse, by doing pair recordings from PV neurons onto immature and mature GC. Those experiments are in our plans but will take time, presumably for a future study. We expanded the subsection “Inhibitory circuits in the control of activity*”* to discuss this topic.

Reviewer 2:

*1) The authors conclude that immature GCs have a wider time window in which excitation can elicit spikes without being affected by inhibition (subsection “Inhibitory control of activity in mature and immature GC”) but it is not clear why the inhibitory conductance fails to provide functional inhibition. Either there is interesting biology that renders immature GCs insensitive to inhibitory conductances, or there is a less interesting (and potentially misleading) possibility that the stimulus paradigm is insufficient to detect the functional consequence of the inhibitory conductance. Even the relatively small conductance shown in*
Figure 3
*in 4wpi cells should provide some degree of inhibition at 1 and 10 Hz*.

Based on the data shown here (Figure 4) and on our previous data (Marin-Burgin et al., Science 2012, Figure Supplement 5) we have demonstrated that the inhibitory conductance fails to provide functional inhibition in 4wpiGC because it is slow, and therefore rises after the time of action potential production. We hypothesize that what is happening here is related to the fact that perisomatic connections onto young GC are immature and thus, functionally different. We clarify this idea in the subsection “Synaptic mechanisms underlying frequency dependence and responsiveness of immature and mature GC”.

*2) In light of the above concern, the authors should carefully consider the use of their stimulus normalization paradigm. This procedure was apparently developed for CA1, where most cells can be driven to spike whereas in the DG this appears not to be the case (the authors suggest that only ∼ 50% of cells spike with inhibition intact). Results from this group's previous study (Supplementary Figure S2E Marin-Burgin; 2012) shows the non-linear relationship between stimulus intensity and % of spiking cells, so it is hard to understand how the pSpike (i.e., the % of spiking cells) provides a sensitive measure of the number of active perforant path fibers. The normalization paradigm shown in*
Figure 1—figure supplement 1
*also does not show that the pSpike is maximal (panel B), and there is a very poor fit of the fEPSP data (panel C). It is not clear why the normalization procedure is required for these experiments, but if it is, the authors should confirm that the fEPSP is linearly related to number of active PP fibers (that could be assayed by the fiber volley) and that their measure of input strength is unaffected by blockade of inhibition. It is important to have similar stimulation parameters to compare the effect of GABAergic inhibition shown in*
Figure 2
*with the results of*
Figure 1*. An alternative approach would be to compare the effect of PTX on spiking in each cell to allow a within-cell comparison, which would obviate the need for the normalization paradigm*.

A) We use normalization for important reasons. Due to variability the way the slice is cut, in the location of the stimulation electrode (although is always in mPP it has some variability), the excitability of a particular slice, it is really impossible to compare across slices using stimulation intensity. We added an example of how different slices present different input output curves of activation in Figure 1—figure supplement 1 to better visualize the importance of normalization. Therefore, we chose 50% fEPSP as a fixed stimulation intensity in order to compare across slices. At that intensity, around 50% of neurons spike. This is the same for CA1 or other regions since the normalization is using the maximum pop spike (were all cells spike) as 100%, and then using the fEPSP recorded at that intensity as the 100% input. An input strength of 50% fEPSP approximately recruits 50% of the cells.

B) We agree that a better measure of the input is the fiber volley. Unfortunately it is not as reliable as the fEPSP in the recordings, in the sense that the signal is not always clearly present since it can be hidden by the stimulation artifact. For that reason we use the fEPSP as an indirect measure of the input. We do that because the fEPSP increases linearly with the fiber volley along a wide range (in the recordings where it was possible to measure it). We have changed the example for one that has incorporated the fiber volley and added an inset panel in Figure 1—figure supplement 1.

C) We always measure the fEPSP in absence of PTX at the beginning of the experiment to have the same input activated in both control and PTX conditions. We did some experiments in the same cells and results were consistent with the data of the pooled slices.

We have clarified the normalization protocol in the Results and in Materials and methods (subsection “Calibration of the input strength for the mPP stimulation”).

*3) The main conclusion is that inhibition dictates the frequency-dependent firing characteristics of mature and 4 wpi cells. But this is not immediately obvious from the data in*
Figure 3*, since the frequency-dependent patterns of excitation and inhibition shown in 3D and E appear to be very similar in mature and 4wpi GCs (despite their different spiking patterns). This could be an issue of presentation – perhaps illustrating frequency-dependent differences in the E/I ratio between mature and 4 wpi cells within the same graph would make this point more clear*.

*In addition, it appears that the synaptic currents in*
Figure 3
*were recorded in AP5 but the spiking data was not. Since the NMDAR component could become a source of depolarization during high frequency stimulation, it should be included in the synaptic analysis or the authors should rule it out by demonstrating a lack of effect of AP5 on spiking*.

A) To better illustrate the frequency-dependent differences in the E/I ratio between mature and 4 wpi cells we added a panel in Figure 3 with the mean ratios and also added a Figure 3—figure supplement 1 with pulse by pulse ratios. In addition, we have included a better explanation of these results in the subsection “Synaptic mechanisms underlying frequency dependence and responsiveness of immature and mature GC*”.*

B) We were also concerned about the effect of AP5 on spiking. We had previously performed experiments controlling for the effect of AP5 on spiking and found no effect (we did not include those experiments in the previous version of the manuscript). We have now mentioned that observation in the Methods, subsection “Evoked postsynaptic currents and conductances”.

*4) The dissection of feed-forward and feedback components of inhibition is potentially very interesting, but I am not convinced by the method since it relies entirely on the selective blockade of mossy fiber synapses by DCG-IV. In fact, 1 micromolar DCG-IV is well known to block medial peforant path evoked EPSCs (i.e.*
[30]
*and many subsequent studies). The authors should repeat this critical control experiment using EPSCs rather than spiking, since supra-threshold excitation could potentially mask a suppression of MPP synaptic strength that would confound all the analysis shown in*
Figure 4*. Furthermore, since GC spiking presumably is required for feedback inhibition, it is difficult to understand how the feedback IPSG is greatest during 40 Hz stimulation, which is the frequency when GCs spike the fewest action potentials (compare*
Figure 1
*with*
Figure 4*). Is there a frequency-dependent recruitment of feedback interneurons? Showing the FB-IPSC during the train might be useful in this regard*.

A) We were aware that DCG4 affects mPP pathway (we added a reference of that in the text now) and for that reason we could not do experiments of the effect of that drug on GC spiking. We used DCG4 only to study feed-forward inhibition, which is also recruited by the mPP pathway. However, inhibitory interneurons tend to display a very low threshold to respond to afferent inputs. Therefore, we reasoned that presumably at the stimulation intensity used in all experiments (50% fEPSP), interneurons would be above threshold even though the drug increased their spiking threshold. Thus, FF-IPSC recorded in GC would not be affected by DCG4. To test this idea we did experiments checking the effect of DCG4 on PV interneurons activation upon stimulation of the mPP (Figure 4—figure supplement 2). Indeed, there was no effect of DCG4 on their spiking at the stimulation intensity that we used for all the experiments. We now better clarify this reasoning in the subsection “Inhibitory circuits involved in the activation of mature and immature GC”.

B) Regarding feedback inhibition, we added a supplementary figure (Figure 4—figure supplement 1) showing pulse to pulse feed-forward and feed-back inhibition traces. As shown in the traces, at 40 Hz the first pulses (that in fact are the only ones evoking spikes at high frequency) can evoke strong feedback inhibition that then decays without growing as the rest of the pulses arrive.

Reviewer 3:

*1) It is not clear how the use of running wheels in all the experiments affects the interpretation of the data. There is growing evidence that suggests that running changes rates of maturation of DGCs and afferent connectivity (**Snyder 2009**,*
*Piatti 2011**,*
*Bergami 2015**). Since one of the aims of this study is to probe differences between immature adult-born DGCs and mature DGCs, it is important that the authors directly address whether running has an effect on activation of 4 weeks old adult-born DGCs following mPP stimulation at different frequency-trains (by comparing with a non-running group)*.

This point is worth clarifying. First, we consider running wheel housing a condition that is slightly closer to the wild environment that normal caging, so animals can voluntarily run. In addition, we use running wheels to increase neurogenesis at the time of viral transduction, which in turn increases the density of labeled cells, as shown in several previous papers from van Praag 1999 onwards. In our labs, we have routinely used wheels for that purpose (Esposito et al. 2005, [26], [25], [33], Marin Burgin et al. 2012, [43]). Even though running can accelerate the speed of maturation at different times of neuronal development, we have demonstrated that 4wpiGC in running animals are still immature (see [33], Marin-Burgin et al. 2012, [43]).

In regard to the mature GC, the answer to this point relies in [26] and [25] papers. There we have compared mature GC generated during development (in sedentary mice) with mature GC born in running adult mice, finding no differences in excitation, inhibition or spiking between those cells.

In the context of those results, our current findings indicate that differences are due to the age of immature GC rather than running conditions. We hope this explanation clarifies the reviewer’s concern.

*2) The notion that intrinsic properties, but local circuitry underlying inhibition and excitations, determines recruitment of immature and mature DGCs in response to PP stimulation is not new and has been addressed previously in Dieni et al.,*
*JNSc 2013**. In this study, the authors used a dual pathway stimulation protocol (MML and OML) and found that ratio of inhibitory and excitatory drive onto DGCs is influenced by stimulus frequency and dictates spiking probability. The relevance of the main findings of this study should be integrated into the discussion of results here*.

We agree that the findings of that study are relevant to ours and we apologize for the missing reference; we have added it in the Introduction and Discussion.

*3) The extent of mossy cell contribution to activation of DGCs following mPP stimulation in slices is not clear. Does stimulation of EC Layer II inputs to DG differentially activate 4 week vs. mature DGCs at 20 Hz? This caveat should be discussed*.

Indeed mossy cells could contribute to excitation and inhibition that GC receive, but we did not address this issue in our work. Nevertheless, we have added a line in the Discussion regarding this topic together with a reference: “Also, there could be a differential contribution of mossy cells, which can affect inhibition and excitation on GC (8)”.